# Endometriosis treatment pathways in the largest private health insurance in Brazil: A real-world data study

Rodrigo Afonso da Silva Sardenberg[1,2], Jose Arnaldo Shiomi da Cruz [1,2,3]*,
Carlos Augusto Lima de Campos[1], Breno Cordeiro Porto[1,3], Kenneth Almeida[1]

1 International Research and Teaching Institute, Hapvida NotreDame, Sao Paulo, São Paulo, Brazil,
2 Ninth of July University, School of Medicine, Sao Bernardo do Campo, São Paulo, Brazil, 3 Federal
University of Sao Paulo, School of Medicine, Sao Paulo, São Paulo, Brazil

* arnaldoshiomi@yahoo.com.br

## Abstract

### Introduction

Endometriosis is a prevalent chronic gynecological condition affecting women of reproductive age, characterized by ectopic endometrial-like tissue growth. Despite its significant impact on quality of life, fertility, and healthcare utilization, data on real-world treatment pathways in Brazil remain scarce.

### Objective

This study explores the clinical and demographic characteristics, diagnostic timelines, and treatment outcomes for women undergoing surgical treatment for endometriosis within Brazil's largest healthcare database.

### Methods

A retrospective cohort study was conducted using data from over 8.8 million lives within the Hapvida NotreDame portfolio. Women undergoing their first surgical procedure for endometriosis between 2006 and 2024 were included. Data on clinical characteristics, healthcare utilization, symptoms, and costs were analyzed.

### Results

The cohort included 5,740 women, with a median surgical age of 37 years. We found that patients with deep endometriosis took longer to undergo surgery than those with superficial endometriosis. Following surgery, there was an increase in the number of normal deliveries and cesarean sections, along with a decrease in the number of emergency gynecological consultations. The median hospital stay for surgery was 1 day and only 66 (1.1%) patients required postoperative ICU care. Additionally, 93 (1.6%) of the patients needed early reoperation, and 239 (4.1%) experienced disease

**Data availability statement:** The data underlying this study cannot be shared publicly due to legal and ethical restrictions related to patient confidentiality and institutional data protection policies imposed by Hapvida NotreDame. Data are available upon reasonable request from the Hapvida NotreDame Research Ethics Committee (Comitê de Ética em Pesquisa – Hapvida NotreDame Intermédica), which is a non-author institutional body responsible for data governance and access approvals. Requests for access to the data can be directed to: Hapvida NotreDame Research Ethics Committee Email: cep@hapvida.com.br.

**Funding:** Funding by Johnson and Johnson.

**Competing interests:** The authors have declared that no competing interests exist.

recurrence and required further surgery. The surgery for patients with deep endometriosis was 40% more expensive than for those with superficial endometriosis, and the cost of diagnostic exams for the disease was equivalent to 69% of the cost of the first surgery for endometriosis.

## Conclusions

This study provides critical insights into the real-world burden of endometriosis, emphasizing the importance of timely diagnosis and surgical intervention. The findings underline the potential for improved quality of life and healthcare utilization through optimized care pathways and resource allocation.

## Introduction

Endometriosis is a gynecological condition characterized by the ectopic growth of endometrial-like tissue outside the uterine cavity, including the peritoneum, mesentery, ovaries, and fallopian tubes [1]. In some cases, the disease deeply infiltrates pelvic organs such as the bowel, bladder, and ureters, characterizing deep endometriosis, which represents the most severe form of the disease. Deep infiltrating endometriosis (DIE) accounts for approximately 20% of all cases and is considered the most severe form of the disease, with peritoneal involvement reported in up to 44% of patients [2]. Affecting approximately 6% to 10% of women of reproductive age, endometriosis ranks among the most prevalent chronic gynecological diseases [3]. The hallmark symptom is cyclic pelvic pain, often accompanied by dysmenorrhea, deep dyspareunia, dysuria, and dyschezia, depending on the organs involved [4]. Although these symptoms are often inappropriately normalized and overlooked due to their chronic nature, they significantly impair quality of life and frequently contribute to psychological conditions, including anxiety and depression [5]. Furthermore, up to 50% of infertile women are diagnosed with endometriosis, underlining its profound impact on hormonal function and fertility [6]. Despite its histological similarities to ectopic endometrium, ectopic lesions exhibit invasive behavior and angiogenic activity, complicating diagnosis and treatment [7]. Surgical treatment is indicated in patients with symptomatic endometriosis refractory to medical therapy, in cases of organ involvement or damage, and in selected patients with persistent infertility [8]. Although surgery may significantly improve symptoms and fertility outcomes, it does not eliminate the underlying disease process, and recurrence remains a relevant clinical concern.[9].

Real-world evidence (RWE) has emerged as a complementary paradigm to traditional randomized controlled trials (RCTs), offering insights derived from real-world data (RWD) collected in clinical practice [10]. Unlike RCTs, which are designed for high internal validity under controlled conditions, RWE captures a broader spectrum of patient experiences, allowing for the examination of outcomes in more diverse populations and real-world settings [11]. This integrative approach not only enhances the applicability of clinical findings but also enables the study of rare diseases and long-term outcomes. By leveraging RWE, healthcare systems can build learning

environments where research is seamlessly integrated with clinical care, addressing gaps left by traditional methodologies and expanding the evidence base for conditions like endometriosis [12].

Studying the patient journey in endometriosis is particularly crucial due to the gaps in understanding key timelines, such as the interval from symptom onset to diagnosis, treatment, and surgical intervention. Notably, the time required for a definitive diagnosis is estimated to take an average of 7 years, highlighting the challenges patients face in receiving timely and accurate care [13]. These delays often exacerbate disease progression and psychological burden, yet data on these critical periods remain scarce in the literature. It is also important to highlight the substantial costs incurred by patients throughout this journey, from the prolonged diagnostic process to the initiation of appropriate treatment. These costs include both direct expenses, such as consultations, diagnostic exams, and surgical interventions, and indirect costs, such as lost productivity, absenteeism, and the overall impact on quality of life [14]. Despite advances in imaging techniques such as transvaginal ultrasound with bowel preparation and magnetic resonance imaging, the diagnosis of endometriosis—particularly deep endometriosis—remains challenging. Variability in imaging expertise, limited access to specialized centers, and the absence of universally adopted classification systems contribute to persistent diagnostic delays.

This study aims to address these knowledge gaps by exploring real-world patient experiences with endometriosis, providing valuable insights into diagnostic and therapeutic timelines. By shedding light on these underexplored aspects, we hope to inform clinical practices and improve outcomes for women suffering from this complex condition.

## Materials and methods

### Study design

This is a population-based, retrospective cohort study, considering the Hapvida Notredame portfolio, which consists of over 8.8 million lives, which accounts for about 4.7% of Brazil population. Hence, this is the largest sample of patients with detailed information in the country. The study received approval from the Institutional Ethics Committee and adheres to the highest standards of research practice as outlined in the Declaration of Helsinki.

This project was assessed and approved by the Brazilian National Ethics on Research Comittee with approval number 7.101.673. Approval in within the submission.

We designed our cohort to encompass all patients that were submitted to surgical treatment for endometriosis. The first surgical procedure for endometriosis was our index event. All information of these patients was extracted from 5 years before de index event to 3 years after the index event.

The first surgical procedure for endometriosis was identified through the authorizations database. There is a national unified table of procedures that identifies all procedures from public and private health systems (TUSS table). We selected all the surgical procedures codes from this table regarding simple endometriosis, deep endometriosis (including bladder and intestinal) and peritoneal endometriosis. Open and laparoscopic approaches were considered. Robotic-assisted approach is unavailable.

### Study variables

After these patients were identified in our datalake by their respective registration number. Our next step was to identify the period each patient was active in the health insurance company. Hence, we identified all activations and deactivations of each patient insurance policies to determine for how long each patient was active on the insurance company. The access in the datalake was performed from October 1st 2024 to December 31st 2024. The extracted data was anonymized to prevent individual participants identification.

We also collected data regarding the age of each patient in the index event, geographical distribution, MRI for endometriosis, number of outpatient consultations, number of emergency room consultations, hospital admissions, intensive care unit admissions and surgical procedures. All these events were extracted with their respective ICD-codes. Data regarding mortality and cause of death was also extracted.

The analyzed surgical procedure reports detailed endometriosis findings as deep endometriosis (characterized by resection of the intestine, colon, or bladder, or explicitly described in the surgical report as deep endometriosis), superficial endometriosis (when findings did not meet the criteria for deep endometriosis), absence of endometriosis, and unavailable information. We considered surgery-related complications as ER consultations and hospital readmissions through ER up to 30 days after endometriosis surgery. Furthermore, no standardized classification system (such as rASRM or ENZIAN) was systematically applied in the surgical reports, as data were derived from routine clinical documentation within a real-world administrative database.

Ethnical data was not explicitly collected for Brazil; however, as a highly multiracial country with data gathered from various regions, the study inherently reflects a diverse representation of ethnic groups.

To analyze patient symptoms, we developed three AI models to extract information related to infertility, dyspareunia, and abdominal pain from patient records. For each symptom, patient records were labeled as "absent," "present," or "unavailable," and a BERT-based algorithm (BioBERTpt-all) was fine-tuned to classify the information into these categories.

The infertility model achieved recall performances of 94.1%, 94.4%, and 93.2% for the "absent," "present," and "unavailable" classes, respectively. The dyspareunia model attained recall rates of 98.5%, 89.7%, and 97.1% for the "absent," "present," and "unavailable" classes, respectively. Lastly, the abdominal pain model achieved recall performances of 97.1%, 96.9%, and 92.8% for the "absent," "present," and "unavailable" classes, respectively.

Data regarding costs of each patient in the Insurance company was also extracted. Only data from 2019 to 2024 was available, hence median and interquartile costs for each treatment step were calculated for estimation. Monetary correction for inflation was performed by the IPCA index (in English, "Broad Consumer Price Index"). The analysis was performed for values of 2024. Values are presented in relative value for preserving company's strategical information.

## Data analysis

Statistical analyses were performed using SPSS software version 29.0 (SPSS Inc., Chicago, IL, USA) and GraphPad Prism version 10.0.0 for Windows (GraphPad Software, Boston, Massachusetts, USA) and a significance level of 5% was considered, meaning results with a p-value of 5% or less ($p \le 0.05$) were considered statistically significant.

## Results

In our cohort, we identified 5740 women that received surgical treatment for endometriosis from 2006 to 2024. Most patients are from the Northeast Region of Brazil (Fig 1), but there are patients in all five regions in Brazil (North, Northeast, Center-West, Southeast and South).

Median age for surgery was 37 (32–42) years-old, with range from 14 to 59 years-old. There were 10 deaths in the cohort period (0,17%). Only one patient had the cause of death endometriosis related but died 67 months after index event. There were 19,714 people-year before index event and 9,574 people-year after index event. The median time of each patient on the insurance company with an active plan was 3.78 (2.28–5.28) years before index event and 1.58 (0.49–2.87) after surgery ($p < 0.001$).

## Symptoms

Infertility and pelvic/abdominal pain were the first assessed symptoms to manifest. Abdominal pain, with infertility reported on outpatient consultations about 19(9–36) months prior to surgery and pelvic or abdominal pain being reported about 18(8–37) months prior to surgery (Fig 2). Dyspareunia was a symptom that appeared significantly later than the others ($p < 0.001$), with its onset about 13 (6–27) months prior to surgery (Fig 2). Patients with deep endometriosis took more time to undergo surgery from endometriosis than patients with superficial endometriosis (Fig 3).

After surgery all symptoms had a significant improvement ($p < 0.001$, Table 1).

| | State | |
|---|---|---|
| 1 | CE | 2592 |
| 2 | BA | 561 |
| 3 | AM | 486 |
| 4 | RN | 375 |
| 5 | PE | 276 |
| 6 | AL | 231 |
| 7 | SP | 210 |
| 8 | MG | 197 |
| 9 | SE | 164 |
| 10 | MA | 139 |
| 11 | PB | 137 |
| 12 | PI | 114 |
| 13 | PR | 111 |
| 14 | PA | 62 |
| 15 | RS | 47 |
| 16 | SC | 30 |
| 17 | RJ | 5 |
| 18 | GO | 4 |
| 19 | DF | 3 |
| 20 | RO | 1 |
| | TOTAL | 5740 |

**Fig 1. Distribution of included patients by Brazilian state.**

## Treatment pathways

Outpatient consultations were discriminated by medical specialty on Table 2-a. The incidence of outpatient consultations in gynecology diminished from 1.64 to 1.42 consultations/people-year, meanwhile the number of outpatient consultations in obstetrics increased from 2.16 to 2.64 consultations/people-year. This behavior was found in deep endometriosis, where gynecology outpatient consultations decreased from 1.97 to 1.70. Obstetrics outpatient consultations increased from 2.50 to 3.01 consultations/people-year. In superficial endometriosis we found the same increase in outpatient consultations in obstetrics, but we also observed a modest numerical increase in gynecology outpatient consultation rates.

Regarding physiotherapy outpatient consultations there were only 208 consultations (1055 consultations/ 100.000 people-year) before index event and 64 consultations (668 consultations/ 100.000 people-year) after index event. Psychology consultations were 6013 (30,501/ 100,000 people-year) before index event and 3,520 (3676/ 100,000 people-year) after index event.

| Symptoms onset (months) | Median (interquartile range) |
|---|---|
| Infertility | 19(9-36) |
| Dyspareunia | 13(6-27) |
| Pelvic or abdominal pain | 18(8-37) |

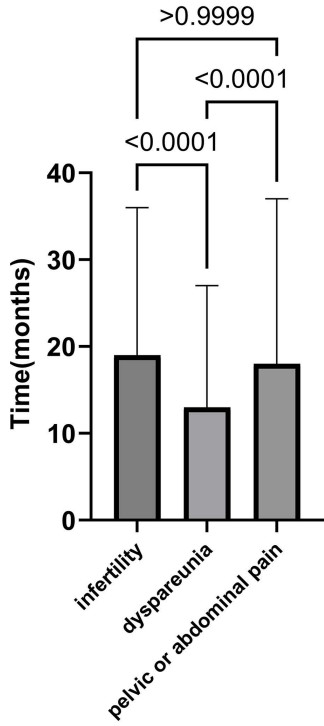

**Fig 2. Time from beginning of symptoms to surgery.**

The incidence of ER consultations (Table 2-b) in gynecology decrease from 0.246 to 0.170 consultations/people-year and increase in obstetrics from 0.258 to 0.811 consultations/people-year.

The incidence of hospital admissions (Table 2-c) for labor increased from 0.27 to 0.44 admissions/people-year. This trend was found in deep and superficial endometriosis.

Regarding the index event, median hospital stay was 1 (1–1) day, with the minimal hospital stay of 1 day AND maximum of 27 days. Sixty-six (1.1%) patients were sent to ICU after surgery, with a median ICU stay of 1.62 (0.62–2.62 days), minimal ICU stay of 1 day and maximum stay of 13 days.

Before index event, patients had a median number of 1(0.5–1.5) procedures. Most common procedures (Table 3) were hysteroscopy, Implantation of Hormonal Intra-Uterine Device and Laparoscopic Cholecystectomy. The rate of Cesarean Section was 557.97/100.000 people-year. Normal Delivery rate was 116.66/100.000 people-year.

After index event, patients had a median number of 1(0.5–1.5) procedures. Most common procedures (Table 4) were hysteroscopy, cesarean section and Laparoscopic surgical treatment of endometriosis. The rate of Cesarean Section was 1984.541/100.000 people-year. Delivery rate was 470.023/100.000 people-year. Two hundred and thirty-nine (4.1%) patients underwent one additional surgery for endometriosis and 3 patients underwent two additional surgeries for endometriosis in our cohort. Findings at index event and operation for recurrence were similar (p=0.901, Table 5). Five cases (0.08%) had elective reoperations within 30 days for findings in the operating room where more complex than preoperative findings and surgical team did not have specialists available for intestinal or ureteral resections.

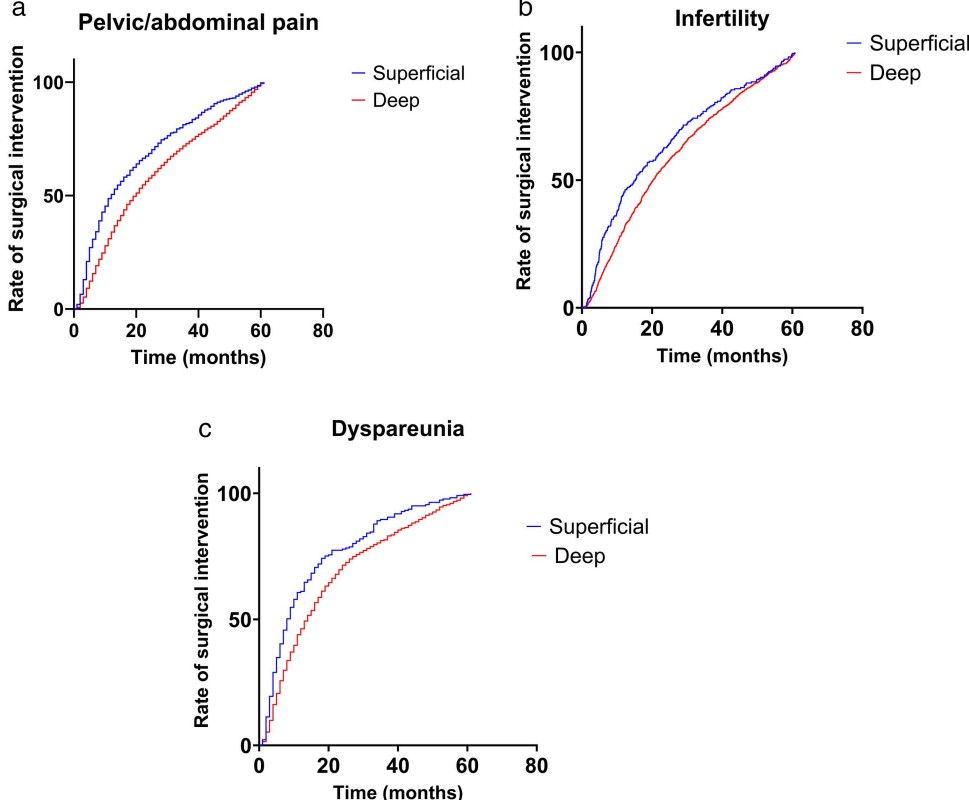

**Fig 3. Relationship between symptoms onset and time for surgical intervention. a.** Comparing the onset of pelvic/abdominal pain in our cohort. Median time for superficial endometriosis was 12 months, and 20 months for deep endometriosis, HR 1.441 (IC 95%, 1,303 − 1,595, p<0.001). **b.** Comparing the onset of infertility in our cohort. Median time for superficial endometriosis was 15.21 months, and 20.38 months for deep endometriosis, HR 1.206 (IC 95%, 1,033 to 1,408, p<0.001). **c.** Comparing the onset of dyspareunia in our cohort. Median time for superficial endometriosis was 8 months, and 14 months for deep endometriosis, HR 1.455 (IC 95%, 1,231 to 1,718, p<0.001).

From the 5740 patients, 93 (1.6%) had urgent reoperations within 30 days from index event. Procedures are discriminated in Table 6. There were 48 cases of reoperation for abdominal obstruction by adherences, 22 cases of persistent urine leakage, 14 fistulas, 6 rectal perforations and 3 ileal perforations. Deep endometriosis had more urgent reoperations and the most severe causes for reoperations in out cohort (p<0.001).

Regarding ER visits (Table 7), the most frequent diagnosis was abdominal and pelvic pain, accounting for 28.34% of consultations (917 cases), followed by encounter for other postprocedural aftercare, representing 21.72% (703 cases), mainly for drain and stitches removal. Other notable diagnoses included endometriosis (5.07%, 164 cases), other disorders of the urinary system (3.34%, 108 cases), and complications of procedures, not elsewhere classified (3.03%, 98 cases). Less common diagnoses included headache (1.92%, 62 cases), dorsalgia (1.89%, 61 cases), other abnormal uterine and vaginal bleeding (1.85%, 60 cases), and cystitis (1.73%, 56 cases). Additionally, 24.65% of consultations (798 cases) fell under the category "others," while 6.46% (209 cases) had no specific information available (Table 8).

## MRI findings

The MRI findings show a significant reduction in cases of deep endometriosis (from 609 to 124) and superficial endometriosis (from 505 to 133) in the post-index period, alongside a proportional increase in cases classified as no endometriosis (from 709 to 472) (p<0.001, Table 9). When compared to surgical findings, MRI demonstrated a sensitivity of 63.7%,

**Table 1. Number of patients presenting each symptom before and after the index surgical event, classified as present, absent, or unavailable in medical records.**

**Outpatient consultations symptoms**

**Dyspareunia**

| | Before index | After index | |
|---|---|---|---|
| Present | 1557 | 435 | |
| Absent | 360 | 104 | |
| No information | 3381 | 4050 | p<0.001 |
| | 5298 | 4589 | |

**Abdominal or pelvic pain**

| | Before index | After index | |
|---|---|---|---|
| Present | 4211 | 1796 | |
| Absent | 747 | 747 | |
| No information | 340 | 2046 | p<0.001 |
| | 5298 | 4589 | |

**Infertility**

| | Before index | After index | |
|---|---|---|---|
| Present | 1279 | 427 | |
| Absent | 2727 | 1883 | |
| No information | 1292 | 2279 | p<0.001 |
| | 5298 | 4589 | |

**Table 2. Evolution of outpatient consultations, emergency room visits, and hospital admissions before and after the index surgical event, expressed as consultations or admissions per person-year.**

| a. Outpatient Consultations (consultations/people-year) | Overall | | Superficial | | Deep | |
|---|---|---|---|---|---|---|
| By medical speciality | Before index event | After index event | Before index event | After index event | Before index event | After index event |
| Gynecology | 1.64 | 1.42 | 0.36 | 0.46 | 1.97 | 1.70 |
| Obstetrics | 2.16 | 2.64 | 0.81 | 1.17 | 2.50 | 3.01 |
| b. ER Consultations (consultations/people-year) | Overall | | Superficial | | Deep | |
| By medical speciality | Before index event | After index event | Before index event | After index event | Before index event | After index event |
| Obstetrics | 0.258 | 0.811 | 0.279 | 0.970 | 0.253 | 0.772 |
| Gynecology | 0.246 | 0.170 | 0.245 | 0.188 | 0.247 | 0.166 |
| c | Overall | | Superficial | | Deep | |
| Hospital admissions (admission/ people-year) | Before index event | After index event | Before index event | After index event | Before index event | After index event |
| Labor | 0.27 | 0.44 | 0.30 | 0.44 | 0.26 | 0.44 |

correctly detecting endometriosis cases, and a specificity of 100%, accurately identifying all cases without endometriosis without generating false positives.

## Costs

The median cost per endometriosis surgery was 22.8% in hospital admission costs, 47.2% in medical team costs and 30% in medications and materials (Table 10). Cases with deep endometriosis had significant more expensive costs in all parameters (p<0.001).

**Table 3. Most common procedures before index event.**

| | Procedures before index event/100,000 people-year |
|---|---|
| Hysteroscopic Surgery for directed biopsy, synechiae lysis, foreign body removal | 2891,346 |
| Implantation of Hormonal Intra-Uterine Device (IUD) | 943,4919 |
| Cholecystectomy Without Cholangiography by Laparoscopy | 532,6164 |
| Hysteroscopy with Resectoscope for myomectomy, polypectomy, metroplasty, endometrectomy, and synechiae resection | 532,6164 |
| Unilateral or Bilateral Oophorectomy or Oophoroplasty | 461,6009 |
| Cesarean Section | 557,9791 |
| Excision of Breast Lesion by Stereotactic Marking or ROLL | 365,2227 |
| Unilateral Ureteroscopic Placement of Double J Stent | 557,9791 |
| Laparoscope with TV and Recording (laparoscopy for surgery) | 309,4248 |
| Cesarean Section (single or multiple fetus) | 299,2797 |
| Normal Delivery | 116,6684 |

**Table 4. Most common procedures after index event.**

| | Procedures after index event/100,000 people-year |
|---|---|
| Hysteroscopic Surgery for directed biopsy, synechiae lysis, foreign body removal | 1608,523 |
| Cesarean Section (single or multiple fetus) | 1984,541 |
| Laparoscopic Surgical Treatment of Peritoneal Endometriosis | 2527,679 |
| Laparoscopic Abdominal Sigmoidectomy | 1180,28 |
| Laparoscope with TV and Recording (laparoscopy for surgery) | 898,2661 |
| Cholecystectomy Without Cholangiography by Laparoscopy | 814,7065 |
| Implantation of Hormonal Intra-Uterine Device (IUD) | 626,6973 |
| Suture of Extensive Wounds, With or Without Debridement | 605,8074 |
| Laparoscopic Unilateral or Bilateral Oophorectomy or Oophoroplasty | 532,6927 |
| Exploratory Laparotomy | 490,9129 |
| Normal Delivery | 470,023 |

Outpatient consultations median cost was 1.0% of the median endometriosis surgery cost, both in deep and superficial endometriosis (p > 0.999). The median cost on exams, including laboratory tests and imaging exams per patient was equivalent to 22% of the overall (deep + superficial) endometriosis surgery cost. When stratified by severity of endometriosis, it accounts for 19.3% of the costs in superficial cases, versus 23.9% in deep endometriosis cases (p < 0.001).

**Table 5. Surgical findings in initial surgery and reoperations for endometriosis.**

| Findings on surgical report | Index event | Reoperation for endometriosis |
|---|---|---|
| Deep endometriosis | 3999 (69.6%) | 156 (65.1%) |
| Superficial | 977 (17%) | 36 (16.3%) |
| No endometriosis | 3 (0.04%) | 0 |
| No information | 761 (13.2%) | 47 (19.6%) |
| Total | 5740 | 239 |
| p=0.901 | | |

**Table 6. Urgent surgical reoperations within the first 30 days after index event, p<0.0001.**

| 30 days urgent reoperations | Superficial | Deep | Overall |
|---|---|---|---|
| Adhrences | 26 (100%) | 22 (32%) | 48 (51.6%) |
| Vesicorraphy | 0 | 22 (32%) | 22 (23.6%) |
| Fistula | 0 | 14 (20%) | 14 (15%) |
| Rectal perfuration | 0 | 6 (10.6%) | 6 (6.5%) |
| Ileal perfuration | 0 | 3 (5.3%) | 3 (3.3%) |
| Total | 26 | 67 | 93 |

**Table 7. ER visits in the first 30 days after index event.**

| 30 days ER visits after index event | |
|---|---|
| Abdominal and pelvic pain | 917 (28.34%) |
| Encounter for other postprocedural aftercare | 703 (21.72%) |
| Endometriosis | 164 (5.07%) |
| Other disorders of urinary system | 108 (3.34%) |
| Complications of procedures, not elsewhere classified | 98 (3.03%) |
| Headache | 62 (1.92%) |
| Dorsalgia | 61 (1.89%) |
| Other abnormal uterine and vaginal bleeding | 60 (1.85%) |
| Cystitis | 56 (1.73%) |
| Others | 798 (24.66%) |
| No information | 209 (6.46%) |
| Total | 3236 |

The median cost per ER visit was divided in 34.2% in unit admission costs, 40.02% in medical team costs and 25% in medications and materials. Patients' with deep endometriosis had more expensive hospital admission costs in ER (p<0.001), but similar costs with medical team, medications and materials. The total ER visit was equivalent to 1.6% of a endometriosis surgery cost (Table 11).

## Discussion

This study aimed to analyze the clinical and demographic characteristics, diagnostic pathways, and treatment outcomes of women with endometriosis, using data from the largest healthcare database in Brazil. The index event, defined as the patient's first surgical intervention for endometriosis, marked a pivotal point for analyzing changes in healthcare utilization.

**Table 8. Rate of ER visits in the first 30 days after index event divided by deep and superficial endometriosis. We took in account 967 patients with deep endometriosis and 218 patients with superficial endometriosis. Chi-square statistics for the absolute number of ER visits was also significative (p = 0.0078).**

| 30 days ER visits after index event | Absolute number | | Relative number | | |
|---|---|---|---|---|---|
| | Deep | Superficial | Deep | Superficial | p |
| Abdominal and pelvic pain | 344 | 76 | 0.356 | 0.349 | 0.843 |
| Endometriosis | 268 | 55 | 0.277 | 0.252 | 0.457 |
| Dorsalgia | 74 | 21 | 0.077 | 0.096 | 0.331 |
| Other abnormal uterine and vaginal bleeding | 111 | 24 | 0.115 | 0.110 | 0.844 |
| Encounter for other postprocedural aftercare | 117 | 5 | 0.121 | 0.023 | **<0.001** |
| Complications of procedures, not elsewhere classified | 59 | 11 | 0.061 | 0.050 | 0.550 |
| Acute upper respiratory infections of multiple and unspecified sites | 52 | 12 | 0.054 | 0.055 | 0.940 |
| Headache | 55 | 5 | 0.057 | 0.023 | **0.039** |
| Others | 705 | 156 | 0.729 | 0.716 | 0.687 |
| Total | 1785 | 365 | 1.846 | 1.674 | – |

**Table 9. MRI findings.**

| MRI | before index | post index |
|---|---|---|
| Deep endometriosis | 609 | 124 |
| Superficial | 505 | 133 |
| No endometriosis | 709 | 472 |
| No information | 1 | 1 |
| Total | 1824 | 730 |
| | | p < 0.001 |

**Table 10. Costs per procedure in relative numbers. The median cost for overall procedures is 100%.**

| | Hospital Admission Costs | Medical team costs | Medications and Materials | Total |
|---|---|---|---|---|
| Superficial | 19.4% | 28% | 19.8% | 67.3% |
| Deep | 23.4% | 52.7% | 33.2% | 109.4% |
| p | <0.001 | <0.001 | <0.001 | <0.001 |
| Superficial + Deep | 22.8% | 47.2% | 30% | 100% |

Results highlighted key shifts, including variations in outpatient consultations, emergency room visits, and surgical procedures. These findings provide a robust foundation for understanding the real-world impact of surgical management on patient trajectories.

Nowadays Endometriosis remains a significant challenge in women's health due to its chronic nature, delayed diagnosis, and complex management [15]. The disease's symptoms, such as cyclic pain, infertility, and psychological distress, are often normalized, delaying treatment and worsening outcomes [16]. The findings of this study underscore the importance of early and tailored interventions, particularly given the observed gaps in healthcare utilization before and after surgical treatment. Literature is pretty well stablished regarding symptoms and fertility improvement after endometriosis surgery, nevertheless we present the first cohort with real world data about endometriosis surgical treatment pathways in the Brazilian supplementary health system [17,18]. In addition, we present an unexpected behavior regarding symptoms (Fig 3): patients with all studied symptoms (pelvic/abdominal pain, infertility or dyspareunia) with deep endometriosis

**Table 11. Costs per procedure in relative numbers. The median cost for overall ER visit is 100%. The median cost for Superficial + Deep Endometriosis ER visits is equivalent to 1.6% of the overall costs for endometriosis surgery.**

| ER visits | Hospital Admission Costs | Medical team costs | Medications and Materials | Total |
|---|---|---|---|---|
| Superficial | 23.7% | 40.04% | 21% | 94% |
| Deep | 34.2% | 40.02% | 26% | 100.7% |
| *p* | *0.06* | *0.15* | *0.29* | *0.0225* |
| Superficial + Deep | 34.2% | 40.02% | 25% | 100% |

underwent surgery significantly later than those with superficial endometriosis. We believe this may occur because these patients take longer from the onset of symptoms to the diagnosis of endometriosis, as deep endometriosis can behave differently or be confused with other differential diagnoses. This finding reinforces the well-documented diagnostic gap in deep endometriosis, in which nonspecific symptoms and limited access to specialized imaging contribute to delayed diagnosis and treatment. Additionally, these patients may require more time for surgery due to the need for more equipment and larger surgical teams, which could lead to a slower pace in conducting their surgeries.

Table 2, shows that the overall number of outpatient consultations in gynecology decreased overall and for deep endometriosis. Nevertheless, obstetricians consultations increased significantly, reflecting indirectly that surgical treatment of endometriosis had a positive impact in patients' fertility. Consultations for endometriosis and other confounding diagnosis decreased significantly after index event, suggesting that operating endometriosis in these women affected positively their quality of life.

At the same time, emergency room consultations rose for obstetrics, indirectly suggesting improvement in fertility. Nevertheless, gynecological ER consultations decreased significantly, suggesting better management and less symptoms from endometriosis. Our findings are in conformation with existing literature [19].

As shown in Table 3, the surgical landscape changed markedly post-index event. The number of cesarean sections increased from 557.97 to 1984.54 per 100,000 person-years, and the rate of normal deliveries rose from 116.66 to 470.02 per 100,000 person-years. Our findings are concordant with current literature, in addition, our report is the first real world data with over six thousand patients in a supplementary health system cohort [20].Conversely, there was a decrease in curettage procedures, likely reflecting improved management strategies and surgical precision. These findings highlight a shift towards minimally invasive approaches, such as laparoscopic surgical treatment for endometriosis, which emerged as a predominant procedure post-index event [21].

Reoperations for recurrence was required in 239 patients (4.1%), it is a value aligned with the current literature. A systematic review from Ianieri et. al. found reoperation rates from 2 to 10% [22]. In total, 93 urgent reoperations were recorded. The most common urgent reoperations within the first 30 days after the index event were related to intestinal adhesions, accounting for 48 cases (51.6%).

Musat et. al reports a rate from 0.1–0.7% of intestinal adhesions for endometriosis; our cohort found a rate of 0.8%. [23]. Vesicorrhaphy was the second most frequent, with 22 cases (23.6%). Darwish et.al. found a rate of 8% of Clavien-Dindo III complications on vesical endometriosis surgery in a 50 patients with urinary endometriosis cohort [22]. The next more frequent complications were fistula repairs in 14 cases (15%). Rectal perforation and ileal perforation with enteric content leakage happened in 9 cases(9.8%). [24]. If we consider only the cases of deep endometriosis on our cohort (3999 patients), our rate of fistulas was 0.3% and our rate of leakages was 0.2%, which seems compatible with current literature. [25] Hudelist et. al. found in his multicentric retrospective cohort encompassing only patients with colorectal deep endometriosis with 937 patients, a leakage rate of 2.03% and a fistula rate of 1.07% [24].

Our MRI findings were also compatible to previous published in the literature. A meta-analysis from Guerriero et. al.[26] found a sensibility of 66% for detection of deep infiltrating endometriosis versus 63.7% in our cohort and a specificity of 97% versus a specificity of 100% in our series. These findings highlight the central role of non-invasive diagnostic tools

in endometriosis care. Improving access to high-quality imaging may reduce diagnostic delays, decrease unnecessary procedures, and ultimately lower the economic burden associated with prolonged diagnostic pathways.

The direct costs of endometriosis management are substantial and encompass a range of healthcare services. Our cohort study reveals significant expenditures across several categories: hospital admissions for surgery (accounted for 22.8% of the whole surgical procedure cost), medical team fees for surgery (accounted for 47.2% of the whole surgical procedure cost), surgical medications and materials (accounted for 30% of the whole surgical procedure cost), outpatient consultations (accounted for 1.0% of the whole surgical procedure cost), diagnostic examinations (accounted for 22% of the whole surgical procedure cost), and emergency room visits (accounted for 1.6% of the whole surgical procedure cost). These figures highlight the considerable financial burden imposed on both patients and healthcare systems. The level of detail we present in our cohort is very valuable and very rarely described in studies. These costs are consistent with findings from other studies. A systematic review by Soliman et al. found that total direct costs ranged from $1109 per patient per year in Canada to $12,118 per patient per year in the USA. The variability observed across studies emphasizes the influence of healthcare systems and service costs on overall expenditure [27]. Furthermore, a study focusing on women with Medicaid insurance in the U.S. reported mean total direct healthcare costs of $13,670 for endometriosis patients, significantly higher than the $5,779 for age-matched controls. This underscores the substantial economic burden, particularly for those with limited insurance coverage [28]. The significant variation in costs across studies necessitates further research to determine factors contributing to these discrepancies. Methodological differences in data collection, cost components considered, and study perspectives contribute to this variability. Our findings unveil a very interesting scenario when assessing deep versus superficial endometriosis: deep endometriosis surgery is about 40% more expensive than superficial endometriosis. Medical team costs are higher, since other surgeons are necessary beyond the gynecologists: urologists, general surgeons and gastrointestinal surgeons are frequently necessary from support. Moreover, more high-cost materials like harmonic scalpels and laparoscopic staplers are required. With more complex procedures, it is not a surprise that hospital admission costs are also higher. To the authors' knowledge, this is the first study focused on discriminating the treatment pathways for deep endometriosis, and the first work to examine the cost difference between these two groups, revealing that deep endometriosis not only involves surgeries that are 40% more expensive.

Costs per visit to ER and outpatient consultations were similar in deep and superficial endometriosis. In addition, costs with imaging and laboratory tests where high in patients with deep endometriosis, probably due to a longer pathway from onset of symptoms to surgery and more expensive exams, like MRIs.

Surgical interventions constitute a major component of direct healthcare costs for endometriosis [29]. Our cohort study reports a median cost per endometriosis surgery divided as 22.8% in hospital admission costs, 47.2% in medical team costs and 30% in medications and materials. The high cost of surgery is further supported by Soliman's findings, which demonstrated that the incremental cost attributed to endometriosis surgeries in the U.S. is substantial [1,27]. However, the cost-effectiveness of surgical intervention remains a subject of debate. A study investigating the surgical removal of superficial peritoneal endometriosis for managing chronic pelvic pain highlighted the need for large, high-quality clinical trials to determine the effectiveness of this approach [30]. The significant recurrence rates after surgery (21.5% within 2 years, 40–50% within 5 years) raise concerns about the long-term cost-effectiveness of surgical treatment alone [31]. The lack of accurate non-invasive biomarkers for endometriosis necessitates laparoscopy for diagnosis in many cases, contributing to the overall cost. The study by Moayeri et al. [32] explored the cost-effectiveness of laparoscopy in unexplained infertility, suggesting that laparoscopy followed by expectant management might be cost-effective under certain circumstances. However, the study's findings were based on a model system and require further validation in real-world settings. Moreover, the potential for postsurgical chronic pelvic pain (CPSP), occurring in approximately 20% of patients within 36 months, further complicates the cost-effectiveness equation. The variation in surgical techniques and the potential for over indication of surgeries also contribute to the overall cost variability and need for improved clinical guidelines [33,34].

Beyond surgical costs, diagnostic procedures and non-surgical management contribute significantly to the economic burden of endometriosis. Our cohort reveals a median cost of 69% of a endometriosis surgery cost per patient for diagnostic exams and an average of 1% for outpatient consultations. The high cost of diagnostic tests is consistent with the observation that a long diagnostic delay is common, often leading to increased healthcare utilization and expenses before a diagnosis is reached. The nonspecific nature of endometriosis symptoms, which overlap with other gynecological and gastrointestinal disorders, contributes to this delay. Surrey et al.[35] demonstrated that longer diagnostic delays are associated with significantly higher pre-diagnosis healthcare costs. The study found that pre-diagnosis costs averaged $21,489, $30,030, and $34,460 for short, intermediate, and long diagnostic delays, respectively, emphasizing the importance of timely and accurate diagnosis. The reliance on invasive procedures like laparoscopy for definitive diagnosis adds to the cost [36]. The development of non-invasive diagnostic tools is crucial for reducing both the diagnostic delay and the overall economic burden [37]. Furthermore, the management of endometriosis-associated pain often involves medications, which contribute to a substantial portion of the overall costs [38]. The use of opioids, while not a standard therapy, is frequent and associated with increased healthcare burden. Alternative, cost-effective pain management strategies, such as complementary and alternative methods, need further investigation [39].

The economic burden of endometriosis extends beyond direct healthcare costs to encompass indirect costs, which significantly impact societal well-being. These indirect costs arise primarily from lost productivity due to pain, fatigue, and other symptoms [40]. Our cohort study does not directly quantify indirect costs, but the high rates of emergency room visits and hospitalizations suggest considerable lost workdays. A study by Armour et al. [40] in Australia found that the majority of costs associated with endometriosis stemmed from reduced productivity, with total economic costs reaching 6.50 billion Int $ per year at a 10% prevalence rate. The impact on women's work life and education is also significant, leading to lost income and reduced economic participation [34]. The study by Culley et al.[41] explored the impact of endometriosis on male partners, revealing that the condition affects several life domains, including sex and intimacy, planning for children, working lives, and household income. Men also reported emotional distress, including helplessness, frustration, and anger. The lack of societal recognition of the impact on male partners results in their marginalization in endometriosis care. These indirect costs are often overlooked but are crucial for understanding the true economic burden of the disease. Moreover, the long-term burden of endometriosis following diagnosis is still understudied, given the chronic nature of the disease and the substantial recurrence of symptoms. Further research is needed to fully quantify the long-term indirect costs and their impact on individuals, families, and society.

Our findings align with prior research highlighting the pivotal role of minimally invasive surgery in the management of endometriosis. For instance, the randomized trial by Daraï et al. demonstrated significant improvements in symptom relief, quality of life, and fertility restoration following surgical intervention [42]. These benefits were observed not only in routine procedures but also in surgeries addressing disease complications, such as colorectal resection for deeply infiltrating endometriosis. Unlike previous studies, our research uniquely leverages real-world evidence to capture a broader spectrum of outcomes, shedding light on long-term trends. These include increased utilization of obstetric care, reflecting the potential reproductive benefits of surgical treatment, as well as a reduction in invasive procedures, indicating a possible shift towards more conservative management strategies in selected cases.

The study has several limitations. As a retrospective cohort study, it is subject to potential biases related to data completeness and coding accuracy. Additionally, even knowing that Brazil is a historically highly mixed country, the absence of ethnic data limits the applicability of findings to specific populations towards the globe. While Table 2 offers a solid foundation for understanding healthcare utilization patterns, a more detailed analysis could better reveal underlying trends. However, this is hindered by the lack of standardization in medical records within the insurance system. Furthermore, the NLP system used to extract symptom data offers a binary classification, which may not fully capture the nuances of symptom progression. An additional limitation is the follow-up period of three years after the index surgical event, which may not fully capture long-term outcomes, recurrence rates, or shifts in healthcare utilization that could emerge over longer observation periods.

It is worth thinking that future studies should consider prospective designs to address data limitations and offer more detailed analyses. Incorporating patient-reported outcomes and integrating biomarkers could provide deeper insights into endometriosis pathogenesis and treatment efficacy. Additionally, exploring reproductive outcomes and their relationship with surgical interventions may guide better reproductive health strategies. Further, cost-effectiveness studies could inform policy decisions and resource allocation for managing this complex condition.

## Conclusion

Our study underscores the significant impact of surgical interventions on improving patient symptoms and reducing the economic burden associated endometriosis. Our findings advocate for the implementation of more effective diagnostic and management strategies to enhance patient care. Moreover, the insights gained from this large-scale study contribute to a better understanding of the real-world dynamics of endometriosis treatment, supporting the development of policies aimed at improving the quality of life for affected women and optimizing healthcare resources.

## Author contributions

**Conceptualization:** Rodrigo Afonso da Silva Sardenberg, Jose Arnaldo Shiomi da Cruz.

**Data curation:** Rodrigo Afonso da Silva Sardenberg, Jose Arnaldo Shiomi da Cruz, Carlos Augusto Lima de Campos, Breno Cordeiro Porto.

**Formal analysis:** Jose Arnaldo Shiomi da Cruz, Carlos Augusto Lima de Campos, Breno Cordeiro Porto.

**Funding acquisition:** Jose Arnaldo Shiomi da Cruz, Carlos Augusto Lima de Campos, Kenneth Almeida.

**Investigation:** Jose Arnaldo Shiomi da Cruz, Carlos Augusto Lima de Campos, Breno Cordeiro Porto.

**Methodology:** Rodrigo Afonso da Silva Sardenberg, Jose Arnaldo Shiomi da Cruz, Carlos Augusto Lima de Campos, Breno Cordeiro Porto.

**Project administration:** Jose Arnaldo Shiomi da Cruz, Breno Cordeiro Porto.

**Resources:** Rodrigo Afonso da Silva Sardenberg, Jose Arnaldo Shiomi da Cruz, Carlos Augusto Lima de Campos, Breno Cordeiro Porto.

**Software:** Jose Arnaldo Shiomi da Cruz, Carlos Augusto Lima de Campos, Breno Cordeiro Porto.

**Supervision:** Jose Arnaldo Shiomi da Cruz, Carlos Augusto Lima de Campos, Breno Cordeiro Porto, Kenneth Almeida.

**Validation:** Rodrigo Afonso da Silva Sardenberg, Jose Arnaldo Shiomi da Cruz, Carlos Augusto Lima de Campos, Breno Cordeiro Porto.

**Visualization:** Jose Arnaldo Shiomi da Cruz, Carlos Augusto Lima de Campos, Breno Cordeiro Porto, Kenneth Almeida.

**Writing – original draft:** Rodrigo Afonso da Silva Sardenberg, Jose Arnaldo Shiomi da Cruz, Kenneth Almeida.

**Writing – review & editing:** Jose Arnaldo Shiomi da Cruz, Kenneth Almeida.

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
