## [Decision Letter · Decision Letter 0]

2 Jan 2026

Endometriosis treatment pathways in the largest private health insurance in Brazil: a real-world data study.

PLOS One

Dear Dr. DA CRUZ,

Thank you for submitting your manuscript to PLOS ONE. After careful consideration, we feel that it has merit but does not fully meet PLOS ONE’s publication criteria as it currently stands. Therefore, we invite you to submit a revised version of the manuscript that addresses the points raised during the review process.

We look forward to receiving your revised manuscript.

Kind regards,

Iwaho Kikuchi, Ph. D., M.D.

Academic Editor

PLOS One

**Journal Requirements:**

“funding by johnson and johnson.”

We note that you received funding from a commercial source: Johnson and Johnson

Please respond by return email with your amended Competing Interests Statement and we will change the online submission form on your behalf.

3. You have indicated that data is available from [ t_jose.cruza@hapvida.com.br ].  Please can we ask you to provide us with a general contact email address for the data requests, so readers can request access in perpetuity. If a general email is not available please provide a link to a website where readers can obtain access to data.].  Please can we ask you to provide us with a general contact email address for the data requests, so readers can request access in perpetuity. If a general email is not available please provide a link to a website where readers can obtain access to data.].  Please can we ask you to provide us with a general contact email address for the data requests, so readers can request access in perpetuity. If a general email is not available please provide a link to a website where readers can obtain access to data.].  Please can we ask you to provide us with a general contact email address for the data requests, so readers can request access in perpetuity. If a general email is not available please provide a link to a website where readers can obtain access to data.

5. We note that Figure 1 in your submission contain map images which may be copyrighted. All PLOS content is published under the Creative Commons Attribution License (CC BY 4.0), which means that the manuscript, images, and Supporting Information files will be freely available online, and any third party is permitted to access, download, copy, distribute, and use these materials in any way, even commercially, with proper attribution. For these reasons, we cannot publish previously copyrighted maps or satellite images created using proprietary data, such as Google software (Google Maps, Street View, and Earth). For more information, see our copyright guidelines: http://journals.plos.org/plosone/s/licenses-and-copyright.

**Additional Editor Comments:**

Both reviewers consider the manuscript worthy of publication. While some revisions are necessary, I am confident that the paper will meet the journal’s standards once the authors have addressed the requested changes.

Reviewers' comments:

Reviewer's Responses to Questions

**Comments to the Author**

1. Is the manuscript technically sound, and do the data support the conclusions?

Reviewer #1: Yes

Reviewer #2: Yes

2. Has the statistical analysis been performed appropriately and rigorously?

Reviewer #1: Yes

Reviewer #2: Yes

3. Have the authors made all data underlying the findings in their manuscript fully available?

Reviewer #1: Yes

Reviewer #2: Yes

4. Is the manuscript presented in an intelligible fashion and written in standard English?

Reviewer #1: Yes

Reviewer #2: Yes

Reviewer #1: This manuscript presents a large-scale study evaluating the economic impact of deep endometriosis. The findings provide valuable insights into the societal burden of endometriosis and help raise awareness of the need for appropriate treatment. I consider the data to be of significant value and worthy of publication.

Reviewer #2: Dear authors,

here are my comments:

- Line 36: the very rare infiltration of lymph nodes is not relevant here. Deep endometriosis with organ infiltration should be mentioned first.

- Line 43: key symptoms: dysmenorrhea, dyspareunia, dysuria, dyschezia should be mentioned.

- Line 51: this is not true. Surgery is the standard in symptomatic endometriosis that can´t be controlled by medial treatment, that causes organ damage or in patients with ongoing infertility.

- Line 53: surgery does not address the underlying disease as well.

- Line 68: ....and treatment.

- Line 70: you could highlight the problems and the current developments in diagnosing endometriosis and imaging.

- Line 118: did any classification play a role here?

- Line 284: diagnostic gap in DE.

- Line 407: role of non invasive diagnosis

Condous G, Gerges B, Thomassin-Naggara I, Becker C, Tomassetti C, Krentel H, van Herendael BJ, Malzoni M, Abrao MS, Saridogan E, Keckstein J, Hudelist G; Intersociety Consensus Group. Non-invasive imaging techniques for diagnosis of pelvic deep endometriosis and endometriosis classification systems: an International Consensus Statement†,‡. Facts Views Vis Obgyn. 2024 Jun;16(2):127-144. doi: 10.52054/FVVO.16.2.012. PMID: 38807551; PMCID: PMC11366111.

- Table 1 is not clear: what do the numbers mean? Contacts? Is the description of the table correct?

- double-check description of table 2.

- Check description of Fig 3B and 3C

- One additional limitation could be mentioned: only 3 years post index event! There might be a shift in results with longer post event period.

.

Reviewer #1: No

Reviewer #2: No

---

## [Author Response · Author response to Decision Letter 1]

4 Mar 2026

Point-by-point response

We sincerely thank the reviewers for their careful evaluation of our manuscript and for the

constructive comments provided. We are pleased that Reviewer #1 recognized the

relevance and value of our data in highlighting the societal and economic burden of deep

endometriosis. We have carefully addressed all suggestions from Reviewer #2, which

substantially improved the clarity, clinical accuracy, and scientific rigor of the manuscript.

Regarding the Introduction, we revised the clinical framing of deep endometriosis to

prioritize organ infiltration, removing the initial emphasis on the rare involvement of lymph

nodes and placing greater focus on bowel, bladder, and ureteral disease. Key symptoms

were updated to reflect standard clinical terminology, including dysmenorrhea, deep

dyspareunia, dysuria, and dyschezia. Statements concerning surgical management were

refined to clarify that surgery is indicated in symptomatic patients refractory to medical

treatment, in cases of organ involvement, or in selected patients with infertility, while also

emphasizing that surgery does not eliminate the underlying disease process and that

recurrence remains possible. Minor semantic adjustments were made to improve accuracy,

including explicit reference to treatment timelines.

We also expanded the Introduction to better highlight current challenges and developments

in the diagnosis of endometriosis, particularly deep endometriosis. A new paragraph was

added addressing limitations in access to specialized imaging, variability in expertise, and

the absence of universally adopted classification systems, thereby contextualizing

diagnostic delays within contemporary clinical practice.

In the Methods section, we clarified that no standardized classification system (such as

rASRM or ENZIAN) was systematically applied, as surgical findings were derived from routine

real-world clinical documentation within an administrative database. This clarification

directly addresses the reviewer’s question regarding classification.

Within the Discussion, we strengthened the interpretation of our findings by explicitly

addressing the diagnostic gap in deep endometriosis. We incorporated a sentence linking

the delayed surgical treatment observed in patients with deep disease to nonspecific

symptom presentation and limited access to specialized imaging. Furthermore, following

the paragraph describing MRI performance, we added a dedicated discussion on the role of

non-invasive diagnostic tools, emphasizing their importance in reducing diagnostic delays,

avoiding unnecessary invasive procedures, and improving cost-effective management. This

section aligns with recent international consensus statements on non-invasive imaging in

deep endometriosis.

All tables and figures were carefully reviewed for clarity. The legend of Table 1 was revised to

clearly describe the meaning of the numbers presented. The description of Table 2 was

corrected to accurately reflect consultation and admission rates per person-year. Figure 3B

and Figure 3C legends were adjusted to correctly correspond to the symptoms analyzed,

without any modification to the underlying data.

Finally, we expanded the Limitations section to explicitly acknowledge that follow-up was

limited to three years after the index surgical event, noting that longer observation periods

may reveal different long-term outcomes, recurrence patterns, or shifts in healthcare

utilization.

We believe that these revisions have substantially strengthened the manuscript and

addressed all reviewer concerns. We are grateful for the insightful feedback and appreciate

the opportunity to improve our work.

Corresponding author:

Jose Arnaldo Shiomi da Cruz, MD, PhD

Cincinato Braga St. 37 - Cj 32

01333-011, São Paulo - SP,

Tel/fax : +55 (11) 3294-5592

Email: arnaldoshiomi@yahoo.com.br

---

## [Editor Report · Decision Letter 1]

6 Mar 2026

Endometriosis treatment pathways in the largest private health insurance in Brazil: a real-world data study.

PONE-D-25-20152R1

Dear Dr. DA CRUZ,

We’re pleased to inform you that your manuscript has been judged scientifically suitable for publication and will be formally accepted for publication once it meets all outstanding technical requirements.

Kind regards,

Iwaho Kikuchi, Ph. D., M.D.

Academic Editor

PLOS One

Additional Editor Comments (optional):

Thank you very much for your careful and thoughtful revisions. I have reviewed your responses and the revised manuscript, and I am satisfied that all reviewer comments have been adequately addressed. The revisions have improved the clarity and scientific rigor of the work.

I am pleased to inform you that your manuscript is now accepted for publication in PLOS ONE.

Thank you again for your valuable contribution and for choosing PLOS ONE for the dissemination of your research.
---

## [Editor Report · Acceptance letter]

PONE-D-25-20152R1

PLOS One

Dear Dr. DA CRUZ,

I'm pleased to inform you that your manuscript has been deemed suitable for publication in PLOS One. Congratulations! Your manuscript is now being handed over to our production team.

Kind regards,

on behalf of

Dr. Iwaho Kikuchi

Academic Editor

PLOS One